# Effect of Functionalization of Texturized Polypropylene Surface by Silanization and HBII-RGD Attachment on Response of Primary Abdominal and Vaginal Fibroblasts

**DOI:** 10.3390/polym16050667

**Published:** 2024-02-29

**Authors:** Maria Teresa Quiles, Alejandra Rodríguez-Contreras, Jordi Guillem-Marti, Miquel Punset, Miguel Sánchez-Soto, Manuel López-Cano, Jordi Sabadell, Janice Velasco, Manuel Armengol, Jose Maria Manero, Maria Antònia Arbós

**Affiliations:** 1General Surgery Research Unit, Vall d’Hebron Research Institute (VHIR), Passeig Vall d’Hebron 119-129, 08035 Barcelona, Spain; manuel.lopezcano@vallhebron.cat (M.L.-C.); jordi.sabadell@vallhebron.cat (J.S.); manuel.armengol@vallhebron.cat (M.A.); 2Department of Basic Sciences, School of Medicine and Health Sciences, Universitat Internacional de Catalunya (UIC), Josep Trueta, s/n, 08195 Sant Cugat del Vallés, Spain; 3Biomaterials, Biomechanics and Tissue Engineering Group (BBT), Department Materials Science and Engineering, Universitat Politècnica de Catalunya-Barcelona Tech (UPC), Escola d’Enginyeria de Barcelona Est (EEBE), Campus Diagonal-Besòs, Av. Eduard Maristany, 16, 08019 Barcelona, Spain; alejandra.maria.rodriguez@upc.edu (A.R.-C.); jordi.guillem.marti@upc.edu (J.G.-M.); miquel.punset@upc.edu (M.P.); jose.maria.manero@upc.edu (J.M.M.); 4Department Materials Science and Engineering, Universitat Politècnica de Catalunya-Barcelona Tech (UPC), Escola d’Enginyeria de Barcelona Est (EEBE), Campus Diagonal-Besòs, Av. D’Eduard Maristany, 16, 08019 Barcelona, Spain; m.sanchez-soto@upc.edu; 5Networking Research Centre of Bioengineering, Biomaterials and Nanomedicine (CIBER-BBN), Institute of Health Carlos III, 28029 Madrid, Spain; 6Abdominal Wall Surgery Unit, Department of General Surgery, Hospital Universitari Vall d’Hebron, Universitat Autònoma de Barcelona (UAB), Passeig Vall d’Hebron 119-129, 08035 Barcelona, Spain; 7Urogynecology and Pelvic Floor Unit, Department of Gynecology, Hospital Universitari Vall d’Hebron, Universitat Autònoma de Barcelona (UAB), Passeig Vall d’Hebron 119-129, 08035 Barcelona, Spain; 8Department of Surgery, Hospital San Rafael, Germanes Hospitalàries, Passeig de la Vall d’Hebron, 107, 08035 Barcelona, Spain; jvelasco.hsrafael@hospitalarias.es; 9Department of General Surgery, Hospital Universitari Vall d’Hebron, Universitat Autònoma de Barcelona (UAB), Passeig Vall d’Hebron 119-129, 08035 Barcelona, Spain

**Keywords:** polypropylene, incisional hernia, pelvic organ prolapse, fibronectin fragment, roughness, functionalization, fibroblast, cytoskeleton, cell adhesion, cell nucleus, surgical mesh

## Abstract

Soft tissue defects, such as incisional hernia or pelvic organ prolapse, are prevalent pathologies characterized by a tissue microenvironment rich in fragile and dysfunctional fibroblasts. Precision medicine could improve their surgical repair, currently based on polymeric materials. Nonetheless, biomaterial-triggered interventions need first a better understanding of the cell-material interfaces that truly consider the patients’ biology. Few tools are available to study the interactions between polymers and dysfunctional soft tissue cells in vitro. Here, we propose polypropylene (PP) as a matrix to create microscale surfaces w/wo functionalization with an HBII-RGD molecule, a fibronectin fragment modified to include an RGD sequence for promoting cell attachment and differentiation. Metal mold surfaces were roughened by shot blasting with aluminum oxide, and polypropylene plates were obtained by injection molding. HBII-RGD was covalently attached by silanization. As a proof of concept, primary abdominal and vaginal wall fasciae fibroblasts from control patients were grown on the new surfaces. Tissue-specific significant differences in cell morphology, early adhesion and cytoskeletal structure were observed. Roughness and biofunctionalization parameters exerted unique and combinatorial effects that need further investigation. We conclude that the proposed model is effective and provides a new framework to inform the design of smart materials for the treatment of clinically compromised tissues.

## 1. Introduction

The connective tissues of the abdominal and vaginal walls share physiological and mechanical properties. Both are altered by injury, disease, stage of life, or surgery, resulting in defects that require advanced biomaterial-based repair. From a surgical perspective, two conditions stand out: incisional hernia (IH) [1] and pelvic organ prolapse (POP) [2]. Their high prevalence and their chronic and disabling nature make them major health problems worldwide [3,4]. For both conditions, risk factors related to the patient, the surgical technique [3,5] or the biological cellular niche [6,7] have been described. However, the dynamics of the underlying cellular and molecular pathomechanisms involved are poorly understood and difficult to study.

Surgical meshes, especially those made of polymers such as polypropylene (PP) [8], are used to repair [9,10] or even prevent [11,12] these defects. Despite their success in many cases, these meshes have several limitations, such as relative biocompatibility [13] and the occurrence of complications and recurrences [14,15]. To address these issues, different materials, designs, manufacturing techniques and different surface treatments have been studied [16,17,18]. Unfortunately, most of the modifications adopted have failed to improve or even worsened clinical outcomes [19], mostly in the case of POP, where the lack of specific preclinical evidence in significant vaginal settings may have contributed to their unsafe reuse from abdominal hernia applications [15]. This emphasizes the need for mesh optimization, recognizing that biocompatibility depends not only on material properties, but also on the dynamic biological plasticity of the patient’s altered connective tissue, which remains in a state of constant metastability [19].

Pioneering research seeks to improve the interactions between material surfaces and adjacent host (patient) tissues, often referred to as the bioactivity zone [20]. To this end, materials are designed to provide bioinstructive cues to influence specific cellular signals and responses [21,22,23]. This approach has shown some promise in hard tissue repair materials [24]. In contrast, the lack of convincing evidence combined with the inherent complexity of manipulating polymers still limits its application to soft tissue repair [25]. To make progress, more appropriate preclinical models are needed, both to study the unique cells of the tissues to be healed and to explore potential new therapies [26,27].

The use of (dysfunctional) primary abdominal and vaginal wall fascial fibroblasts may be useful for this purpose. Specific changes in the native cellular niche have been reported in IH [28,29] and POP [5], including: extracellular matrix (ECM) alterations, tissue atrophy, deregulated inflammatory signaling and macromolecular synthesis and degradation, and changes in mechanical properties. These changes affect the phenotype of local fibroblasts, which exhibit structural and functional abnormalities, as well as characteristics of cellular fragility and cell death induction [30,31,32]. In addition, differences in the abdominal and vaginal biological niches are likely to contribute to differential responses to PP mesh and consequently the extent to which the mesh can integrate [15]. Innovative polymer-based interventions may offer streamlined, reproducible strategies to enhance cellular function and promote stable interfaces. Tailoring surgical meshes to the characteristics of complex (dysfunctional) target tissues and their anatomical environment can improve biocompatibility and reduce inflammation and foreign body reactions. In addition, incorporating bioactive substances or growth factors into mesh materials can stimulate tissue regeneration, accelerate healing and reduce mesh-related complications.

PP is one of the major polymers used in surgical procedures for IH and POP repair. In general, surgical meshes available for IH and POP repair fall into two main categories: non-resorbable and partially or fully resorbable prostheses. Examples of non-resorbable materials include PP, polyester, expanded polytetrafluoroethylene (ePTFE), polyvinylidene fluoride (PVDF), and polyethylene terephthalate (PET). On the other hand, partially or fully resorbable prostheses can be synthetic (e.g., polyglycolic acid (PGA), polylactic acid (PLA), poly-4-hydroxybutyrate (P4HB)) or biological in nature. These meshes can be constructed from a single material or as composite structures. In addition, ongoing research addresses the development of new engineered materials, such as hydrogels or electrospun fiber meshes, with the goal of improving the safety and efficacy of these repair procedures, as highlighted in recent reviews [33,34,35]. The superiority of PP mesh in IH and POP repair is due to its combination of properties, including biocompatibility, strength, durability, minimal tissue reaction, ease of handling, cost effectiveness, proven efficacy, wide availability and versatility. In comparison, materials such as PET, PVDF, and ePTFE offer improved biocompatibility but may lack the mechanical strength of PP. Resorbable materials, while offering advantages in biodegradability, require additional considerations regarding their degradation kinetics and the provision of mechanical support over time.

Since the initial response of the host (patient) environment to mesh depends significantly on the surface properties of the materials, here we created PP disks, focusing on two key surface modifications: microscale roughness variation and chemical variation with the covalent attachment of an adhesion-promoting molecule. The disks were designed to allow these modifications to be studied individually and in combination, and compared to bare surfaces. As a proof of concept, we investigated how these modified PP surfaces interact with human fibroblasts derived from abdominal and vaginal wall tissues of control patients, in terms of cell adhesion, cytoskeletal response and nuclear morphology. 

We chose micrometer-scale features because they are comparable in size to the cell body and result in whole-cell contact guidance effects that influence cellular properties such as cell morphology and migration [36]. Another important aspect is that PP does not promote cellular adhesion due to its lack of an adhesion motif. Although several modifications have been performed to improve cell attachment to PP devices [37,38,39,40,41], to our knowledge, none have incorporated ECM-derived proteins, fragments, or peptides containing cell-adhesive sequences such as arginylglycylaspartic acid (RGD). Among the various possible ECM molecules, in previous studies we generated a recombinant fragment spanning the type III12-14 (heparin-binding II, HBII) repetitions of fibronectin (a protein strongly involved in tissue regeneration), which has been shown to induce bone differentiation in mesenchymal stem cells due to its ability to bind growth factors [42]. The fragment has been mutated to include an RGD sequence (HBII-RGD), which promotes both cell adhesion and differentiation/activation properties when osteoblasts [43] or fibroblasts [44] are applied to solid materials.

While acknowledging the inherent reductionism of in vitro models (including the limitations of fully recapitulating in vivo tissue conditions and the complex challenge of precisely tuning biophysical parameters for optimal integration into scaffolds), we believe that by coupling microroughness with biochemical modification combined with clinical material holds promise to advance our understanding of tissue-material interactions in diverse physiopathological scenarios. Through this exploration, we aim to contribute to the development of improved and safer materials and to refine strategies for predicting clinical outcomes with greater accuracy. 

## 2. Experimental Section

### 2.1. Material

Polypropylene (PP) pellets were provided by B. Braun (Melsungen, Germany). All reagents were purchased from Sigma-Aldrich (Merck KGaA, Darmstadt, Germany). Cell culture media and supplements were purchased from ThermoFisher Scientific (Waltham, MA, USA).

### 2.2. Sample Preparation

Polypropylene (PP) pellets were processed into plaques (9.8 × 9.8 cm and 3 mm in height) by injection molding under the following conditions: 110 Tn closing force, 245 °C injection temperature, 1000 bar pressure, 200 cm^3^/s injection speed. Smooth PP plates were used as controls.

In order to ensure reproducibility of the surface roughness (Ra), controllable Ra molds were prepared. 

A textured mold with two different micro-roughness levels was prepared using a sandblasting process with aluminum oxide (corundum) particles with an average grain size according to FEA Standard F of 63 and 120 µm (F220 and F120, respectively; MPA, Cornellà, Spain) [45] and applying 4-bar shot pressure at an angle of incidence of 60 degrees. Textured PP plaques were then obtained by injection molding in the texturizer molds under the same conditions as for the preparation of smooth plaques. 

Subsequently, 10 mm diameter disks were obtained from both smooth and textured plaques by subtractive machining (turning), protecting the surface of interest to preserve the roughness obtained. The samples were cleaned by sequential ultrasonic immersion baths in ethanol and acetone for 5 min each, dried with nitrogen, and stored until use.

### 2.3. Synthesis of Recombinant HBII-RGD Fragments

The HBII-RGD fragment was produced by standard recombinant DNA methods as previously described [43,44]. Briefly, the DNA fragment of HBII was inserted into the pGEX-6-P1 plasmid (GE Healthcare, Buckinghamshire, UK) and transformed into DH5α cells (Invitrogen, Carlsbad, CA, USA). After sequencing, correct plasmids were mutated in two rounds to introduce an RGD sequence from a PGV sequence using the QuickChange Lightning site-directed mutagenesis kit (Agilent Technologies, Santa Clara, CA, USA). Mutations were performed in a T100 thermal cycler (Bio-Rad, Hercules, CA, USA) according to the manufacturer’s instructions. Primers for the P233R mutation were 5′-CT CGG CCC CGC CGT GGT GTC ACA GA-3′ (forward) and 5′-TC TGT GAC ACC ACG GCG GGG CCG AG-3′ (reverse), and the primers for the V235D mutation were 5′-CC CGC CGT GGT GAC ACA GAG GCT AC-3′ (forward) and 5′-GT AGC CTC TGT GTC ACC ACG GCG GG-3′ (reverse). After each mutation, plasmids were inserted into DH5α cells, purified and sequenced. The correct constructs were transformed into BL21 cells (New England BioLabs, Hitchin, UK) and the resulting colonies were grown in LB at 37 °C with continuous shaking. Cells were induced by the addition of 1 mM isopropyl-β-D-1-thiogalactopyranoside (IPTG) for 4 h at 37 °C and centrifuged. After sonicating the cells, HBII-RGD was purified using a GSTrap affinity column, with on-column removal of the GST-tag. 

### 2.4. PP Functionalization by Silanization and HBII-RGD Covalent Attachment

#### 2.4.1. Oxygen Plasma

Oxygen plasma was used to activate PP surfaces. A low-pressure plasma system (12 MHz in a Femto, Diener Electronic, Ebhausen, Germany) was used to evaluate different oxygen plasma exposition times (1, 2, and 4 min). Contact angle measurements were conducted to determine the surface wettability and to select the optimal plasma exposition time.

#### 2.4.2. Silanization Process

The HBII-RGD fragment was covalently attached to PP discs by silanization as carefully described in previous studies [43,44]. Briefly, PP discs were cleaned and activated by oxygen plasma for 2 min at 12 MHz in a Femto low-pressure plasma system (Diener Electronic, Germany). After surface activation, the samples were immersed in 0.08 M solution of (3-aminopropyl)triethoxysilane (APTES, Sigma-Aldrich, St. Louis, MO, USA) at 70 °C for 1 h, rinsed with different solvents, and crosslinked with 7.5 mM solution of N-succinimidyl-3-maleimidepropionate (SMP). Finally, the HBII-RGD fragment was immobilized on the PP surface at a concentration of 100 μg/mL in phosphate-buffered saline (PBS) to ensure surface saturation.

### 2.5. Surface Characterization

#### 2.5.1. Field Emission Scanning Electron Microscopy (FESEM)

Topography of the PP surfaces was evaluated by a compact Phenom XL Desktop SEM (PhenomWorld, Eindhoven, The Netherlands) with a Color CCD optical camera (magnification range 3–16×) operating at 15 kV working voltage to examine PP surfaces with micro-textures. Energy Dispersive X-ray spectroscopy (EDS) analyses were used to determine the presence of any contamination on the surfaces. All samples were Pt/Pd coated prior to electron microscopy observation. 

#### 2.5.2. Confocal Microscopy

The arithmetic mean roughness (Ra), maximum profile height (Rz) and root mean square roughness (Rq) parameters were analyzed using a laser scanning confocal microscope, model OLS Olympus Lext 3000 (Olympus Corporation, Shinjuku, Tokyo, Japan). Images were taken at 1000 magnifications for the measurement of the roughness parameters.

#### 2.5.3. Wettability and Contact Angle (CA)

Water contact angles were measured to characterize the wettability and hydrophilicity of PP surfaces. Static contact angles were estimated using the sessile drop method (Contact Angle System OCA15 Plus; DataPhysics Instruments, Filderstadt, Germany). Measurements were made in triplicate for three samples at room temperature, with a volume of 3 μL, and a dose of 1 μL/s. Results were analyzed using SCA 20 software (DataPhysics Instruments, Filderstadt, Germany).

#### 2.5.4. X-ray Photoelectron Spectroscopy (XPS)

XPS was used to analyze the chemical composition of the treated PP surfaces. XPS spectra were acquired with an XR50 Mg anode source operating at 150 W and a Phoibos 150 MCD-9 detector (D8 advance, SPECS Surface Nano Analysis GmbH, Berlin, Germany). High resolution spectra were recorded with a pass energy of 25 eV at 0.1 eV steps and pressure below 7.5 × 10^−9^ mbar. Depending on the sample, the binding energies were referred to C1s, O1s, N1s and Si 2p signals. Two samples were analyzed for each working condition.

### 2.6. Cell Response

#### 2.6.1. Isolation and Culture of Primary Human Fibroblasts

The study was approved by the Ethics Committee of Vall d’Hebron University Hospital (project number PI17/01236), and was conducted in accordance with the Declaration of Helsinki. Informed written consent was obtained from all patients before their participation in the study. 

Primary abdominal wall fibroblasts were obtained from biopsies (~50–100 mg) taken from the abdominal fascia of volunteer donors undergoing elective abdominal surgery at the Department of General Surgery, Vall d’Hebron University Hospital (Barcelona, Spain). Patients with a history of abdominal surgery, diabetes mellitus, obesity, or connective or systemic inflammatory diseases were excluded. Primary vaginal fibroblasts were isolated from the fibromuscular layer of vaginal wall biopsies (~100–200 mg) obtained from women undergoing hysterectomy for benign conditions at the Department of Gynecology of the same hospital. Women with a history of connective tissue disorders, endometriosis, prior pelvic reconstructive surgery, or cancer were excluded.

Both tissues were processed similarly. Briefly, according to an established protocol [30] each biopsy was finely minced and digested with trypsin (0.625%; *w*/*v*) and EDTA (0.02%; *w*/*v*) in D-PBS in a two-step process that included an initial 24-h incubation at 4 °C, followed by a second 45-min incubation at 37 °C after trypsin replacement. Supernatants from each digestion step, as well as the residual tissue from each biopsy, were washed by centrifugation and plated in DMEM supplemented with penicillin (100 units per mL), streptomycin (100 μg mL^−1^), L-glutamine (2 mM), and 10% fetal bovine serum at 37 °C in an atmosphere containing 5% CO_2_. Fibroblasts were then expanded and experiments performed at cell passages 5 and 6. Three cell lines were developed from each tissue type (abdominal wall fascia and vaginal wall).

#### 2.6.2. Fibroblast Cell Culture on Modified PP Substrates

Polypropylene discs were sterilized using a Sterrad^®^ hydrogen peroxide gas plasma sterilizer (Advanced Sterilization Products; Irvine, CA, USA), placed in sterile 12-well cell culture plates (Corning Costar; Fisher Scientific, Madrid, Spain), and immobilized using CellCrown™ inserts (Merck, Darmstadt, Germany) to provide a standardized environment for the cells (Figure 1a). The discs were washed twice with sterile D-PBS under aseptic conditions, and 1.5 mL of standard growth medium was added to each well. Fibroblasts were then seeded onto the PP surfaces at a concentration of 3 × 10^4^ cells/disc into the CellCrown™ inserts as illustrated in Figure 1b. After overnight incubation (37 °C, 5% CO_2_), the growth medium was carefully replaced and fibroblasts were incubated for either an additional 24 h or 5 d, with medium renewal on day 3. To monitor cell behavior, two control wells without PP discs were included in all experiments. The control wells allowed evaluation of cell growth, as the opaque nature of PP discs prevented microscopic observation. At the end of the incubation periods, fibroblasts were gently washed with pre-warmed D-PBS and used for subsequent analysis.

#### 2.6.3. Evaluation of Human Fibroblast Adhesion and Viability on PP Substrates

Fibroblast adhesion and viability on the experimental PP surfaces were evaluated indirectly by crystal violet staining of adherent cells 48 h and 6 d after seeding [46]. We performed the first evaluation 48 h after seeding of fibroblasts because previous tests in our laboratory indicated that cell adhesion was slower in the early phase of adaptation to PP than to standard culture plates. After washing with D-PBS, cells were incubated for 20 min at room temperature (RT) in a solution containing 0.1% (*w*/*v*) crystal violet, 1% formaldehyde, and 1% methanol in D-PBS. After rinsing with tap water, fibroblasts were allowed to dry overnight.

Micrographs at 10× magnification were taken from four equidistant points on the central part of the PP surface using a digital camera (Nikon DS-Ri2, Nikon Europe B.V.; Amstelveen, The Netherlands) coupled to an inverted microscope (Nikon Eclipse Ts2R). The percentage of the image area covered by cells and the associated intensity were determined from the threshold and intensity inverted regions using ImageJ software (v1.54f, National Institutes of Health, Bethesda, MD, USA) as previously described [30]. Two replicate discs per condition were analyzed for each fibroblast line.

#### 2.6.4. Immunofluorescence

Fibroblasts were cultured on PP surfaces for 48 h and processed for immunofluorescence using the Actin Cytoskeleton/Focal Adhesion Staining Kit (FAK100; Sigma-Aldrich; Merck, Darmstadt, Germany) according to the manufacturer’s instructions. Cells were fixed in 4% paraformaldehyde in D-PBS for 20 min at room temperature (RT), washed three times in D-PBS, and permeabilized in 0.1% Triton X-100 for 5 min. The cells were then blocked in 1% bovine serum albumin (BSA) for 30 min. Both primary mouse monoclonal anti-vinculin (1:100; clone VIIF9; Sigma-Aldrich) and secondary Alexa Fluor 488-conjugated goat anti-mouse IgG1 (1:400; Thermo Fisher Scientific, Waltham, MA, USA) antibodies were then applied to the cells and incubated at RT for 60 min. TRITC-conjugated phalloidin (1:1000; Sigma-Aldrich) was added together with the secondary antibody solution. Nuclear DNA was visualized using 4′,6′-diamidino-2-phenylindole dihydrochloride (DAPI 1:1000; Sigma-Aldrich) for 5 min. Cells on substrates were mounted in ProLong Diamond Antifade Mountant (Pierce, Thermo Fisher Scientific) and covered with 24 × 60 mm coverslips to facilitate laser scanning confocal microscopy (LSCM). All samples were allowed to harden for at least 24 h prior to imaging. Negative control staining in the absence of primary antibody was performed on cells cultured on 18 mm round glass coverslips. Two independent cultures were imaged for each experimental set up and two replicates per condition were analyzed.

#### 2.6.5. Image Acquisition and Quantitative Analysis

Immunostained cells were observed using a laser scanning confocal microscope LSM980 (Carl Zeiss, Jena, Germany) with excitation wavelengths of 405, 488 and 561 nm and magnifications of 10× and 40×. The discs were imaged using the same laser intensity settings during image acquisition. Images at 40× were acquired at 0.207 × 0.207 × 0.300 p × µm^−1^ (xyz, respectively). Acquisitions in the Z plane were optimally sampled (Nyquist–Shannon theorem) in sequential scan settings to avoid specific crosstalk between different fluorophores and to achieve the best signal-to-noise ratio during the process. Post-acquisition image deconvolution was performed using Zeiss Zen software (v. 3.3, blue edition, Carl Zeiss, Jena, Germany), and maximum projections of image stacks were generated for presentation.

Z-stacks obtained from deconvolution microscopy were processed using the ImageJ software (NIH). Channels in each Z-stack were split and Z-projections were generated using the sum-slices method for image quantification. The TRITC and AlexaFluor 488 channels were binarized using Huang’s fuzzy thresholding method to create a mask of either F-actin or vinculin staining. The integrated density was then measured within the area of interest. At least two random images per disc were acquired for quantification.

Nuclear morphology was measured using the analysis component of the NIH Image J software. Similarly, the Z-projection of the DAPI/blue channel was binarized by thresholding. Morphologic nuclear features were then determined from individual segmented nuclei, including area, circularity, and aspect ratio.

### 2.7. Statistics

Material characterization measurements were expressed as the mean with standard deviation. The *t*-test with a 95% confidence interval was used to evaluate the statistical differences of the parametric data in the mean values between the two groups.

For the cell response study, statistical data analysis was performed using GraphPad Prism 6.0 (GraphPad Software Inc., San Diego, CA, USA). D’Agostino-Pearson and Shapiro–Wilk normality tests were used. For each experimental determination, multiple comparison tests (two-way ANOVA with Holm-Sidak’s correction, or Kruskal-Wallis test with Dunn’s correction) were used to assess statistical differences between the surfaces with different roughness, functionalized or not with HBII-RGD, compared to the respective controls. Statistical significance was represented by * *p* < 0.05, ** *p* < 0.01, *** *p* < 0.001 and ns = no significance.

## 3. Results

### 3.1. Micro-Topography

Two homogeneous micro-topographies were generated on PP surfaces during the injection molding process using Ra-controlled molds. 

A fine texture with roughness of 1.62 ± 0.14 µm and a gross roughness of 4.16 ± 1.38 µm were obtained. The surfaces examined under confocal and electron microscopes and analyzed by EDS showed consistent topographies with no signs of contamination (Figure 2).

### 3.2. Oxygen Plasma Activation Process

The application of oxygen plasma to PP is related to the formation of OH groups and leads to an increase in wettability. Therefore, the contact angle of the samples was determined after different exposition times to oxygen plasma. After 2 min of plasma treatment, the contact angle changed significantly and the hydrophobicity of the surfaces increased. Therefore, 2 min was chosen as the treatment time prior to silanization.

### 3.3. Functionalization of SMOOTH PP Surface: HBII-RGD Covalent Attachment via Silanization

In order to monitor the reaction, XPS and contact angle analyses were performed on samples during the different steps of the silanization process. Figure 3a shows the functionalization steps of the PP surface: oxygen plasma activation and functionalization with APTES molecule. The contact angle decreased between the untreated PP and the plasma activated PP. No significant differences in surface wettability were observed between the different steps of the functionalization process (Figure 3b).

The results of the XPS analyses are summarized in Figure 3c and Table 1 and Table 2. The atomic percentage analyses of the control sample showed a composition mainly based on C and a small amount of O. In the PP control, the C element was mainly combined with C-C and C-H, which is the chemical structure of the backbone of PP. The oxygen present was combined with O-C, O-H and O-C=O, most likely in the form of hydroxylic and carboxylic groups at the end of the polymer chain. An increase in the percentage of O was observed in the PP sample treated with oxygen plasma (2 min) in the form of -C-OH and C=O, increasing the hydroxylic groups and forming new ketone groups. 

The samples silanized with APTES showed the presence of Si and N in the form of Si-O and amine groups, respectively. This result confirms that the first step of the silanization process was carried out correctly, as the silane was forming a bond with the OH groups on the plasma-activated PP surface. Additionally, the presence of APTES is demonstrated by the increase in the N signal, as APTES contains amine groups. When the HBII-RGD fragment was added, we observed a further increase in the N-signal due to the presence of amine groups in the protein. This confirms the correct functionalization of the HBII-RGD on the PP surface.

### 3.4. Functionalization of ROUGH PP Surface: HBII-RGD Covalent Attachment via Silanization

The contact angle of the samples was evaluated before and after the silanization process (Figure 4). The relationship between roughness and wettability was defined by Wenzel in 1936, who stated that an increase in surface roughness would enhance wettability caused by the surface chemistry [47]. However, adding surface roughness tends to further increase hydrophobicity when the surface is chemically hydrophobic, as in the case of PP. Thus, in our case, wettability decreased with increasing roughness. However, the silanization process increased the wettability of PP surfaces, indicating a chemical change when functionalization with the HBII-RGD fragment was completed.

Results from the XPS analyses are summarized in Figure 4b,c together with Table 3. The untreated smooth PP surfaces, used as a control, showed the same results as those commented on in the previous section. The presence of Si was almost negligible for the control, but slightly increased on the treated surfaces due to the residual alkoxysilanes (APTES) of the functionalization process. When HBII-RGD was added, the presence of N and O increased in both high and low roughness PP surfaces. The percentage of N was 11.85 ± 2.028% and 10.283 ± 5.597% on high and low roughness PP surfaces, respectively. In both samples, all N was present in the form of N-(C=O)-, indicating the presence of HBII-RGD on PP surfaces.

### 3.5. VW and AW Fibroblast Response to the Experimental PP Surfaces

#### 3.5.1. Fibroblast Adhesion and Viability

We first confirmed by crystal violet staining that all the experimental surfaces supported fibroblast attachment, spreading, and growth for 6 days after seeding, irrespective of the fibroblast origin. However, significant differences were observed 48 h after seeding, depending on the specific PP surface modification analyzed, as summarized in Figure 5 and Figure 6.

Representative microscopic images of the top surface of PP discs with crystal violet stained fibroblasts showed that microtopographies significantly reduced cellularity for both AW and VW fibroblasts compared to smooth samples (56% and 65%, respectively) 48 h after seeding, especially when Ra was 4.16 µm (Figure 5b and Figure 6b). Cellular clumps and debris were also observed on these surfaces. Figure 5c and Figure 6c show how functionalization of the PP surfaces with HBII-RGD promoted cell adhesion under certain conditions compared to the untreated substrates. A significant twofold increase in cellularity was observed on functionalized smooth surfaces seeded with AW fibroblasts compared to non-functionalized surfaces. To a lesser extent, the same effect was observed with VW cells (1.21-fold increase). However, when HBII-RGD was applied to rough PP, no significant influence on cellularity was observed, except for the 4.16-µm Ra surface cultured with AW fibroblasts, where cell adhesion was significantly enhanced and cellular clumping was reduced compared to non-functionalized PP. Notably, AW fibroblasts cultured on functionalized rough substrates also exhibited a more elongated morphology with extending projections compared to their counterparts (Figure 5b). Consistent with these results, two-way ANOVA analysis revealed a significant interaction between roughness and HBII-RGD, particularly for AW fibroblasts (*p* < 0.0001) versus VW fibroblasts (*p* < 0.01). At day 6, AW fibroblasts cultured on non-functionalized 4.16 µm Ra surfaces showed the lowest level of cellularity (~32% relative to smooth PP; shown in Figure 5c and Figure 6c). 

#### 3.5.2. Fluorescent F-Actin and Vinculin Staining

We next explored the impact of PP surface modifications on fibroblast cytoskeletal phenotypes by comparing the Z-slices of confocal images after staining cells for vinculin, F-actin, and DNA. Figure 7a and Figure 8a show the maximum intensity projections of stained AW and VW fibroblasts cultured on the experimental surfaces for 48 h along with orthogonal Z-axis micrographs. Although both fibroblast types demonstrated adhesion to all substrates as evidenced by crystal violet staining, our findings indicate that the origin of fibroblasts played a critical role in their response to PP in terms of changes in cell morphology and cytoskeleton. AW fibroblasts showed numerous well-defined actin fibers and better spreading on substrates compared to VW fibroblasts (Figure 7a), which exhibited thinner and more compressed actin fibers (Figure 8a). Vinculin staining was diffuse in all cells with no clear focal adhesion spots, but its intensity changed with surface modifications, as shown in Figure 7b and Figure 8b. A significant increase in the staining intensity of actin microfilaments and vinculin in AW fibroblasts (~2-fold and 2.8-fold, respectively, when Ra was 4.16 µm) induced by PP surface roughness is shown in Figure 7b. Functionalization of these surfaces with HBII-RGD slightly enhanced F-actin intensity in these cells. In contrast, Figure 8a shows that VW fibroblasts seeded on non-functionalized PP surfaces exhibited a distinct adhesion profile, forming tightly packed clusters of cells that increased in number and size with surface Ra (Figure 8c). The orthogonal Z-axis micrographs shown in Figure 8a help to identify these clusters by showing their thickness. The slight increase in fluorescence intensity of F-actin and vinculin observed in VW fibroblasts, especially vinculin in the non-functionalized surface with 1.62-µm Ra, was likely influenced by cell clustering and compression (Figure 8b). Interestingly, the presence of HBII-RGD reduced cell clumping, as shown in the images in Figure 8a. Furthermore, this effect was associated with a decrease in the fluorescence intensity of both F-actin and vinculin (Figure 8b).

#### 3.5.3. Nuclear Morphology

Having determined that PP surface treatments can differentially affect the organization and structure of the cytoskeleton in AW and VW fibroblasts, we wanted to assess whether PP microtopographies and functionalization could alter nuclear morphology. Figure 9 shows that nuclear morphological measurements differed significantly between AW and VW fibroblasts. Vaginal wall fibroblasts exhibited smaller nuclei and high variability in nuclear circularity and aspect ratio compared to AW cells. Surface roughness was found to be the primary factor influencing nuclear morphology, while functionalization had a minimal impact. The effects on nuclear morphology differed between AW and VW cells. A surface Ra of 1.62 µm caused an increase in the nuclear area of AW fibroblasts, whereas it reverted to smaller values when cultured on 4.16 µm Ra. Microtopographies, in turn, led to a significant 25% decrease in the nuclear area of VW fibroblasts, accompanied by an increase in nuclear heterogeneity, all associated with the formation of cell clumps (Figure 8c).

## 4. Discussion

The present study investigated the response of human abdominal and vaginal wall fibroblasts to surface-modified PP materials by measuring cell adhesion in terms of cellularity and characterizing cytoskeletal and nuclear structure in material-adherent cells. We used a multifaceted approach to surface modification, including microscale roughness, functionalization, and a combination of both strategies. 

Polypropylene is one of the most commonly used materials for vaginal and abdominal wall surgical repair. However, its effectiveness is limited by complications related to tissue integration. Many questions remain regarding the precise mechanisms governing this process, particularly those related to cell-substrate interactions, which are critical to biomaterial integration. Studies have shown that cell behavior can be influenced by both topographical and biochemical cues that regulate a wide range of cell functions, including adhesion, proliferation, morphology, or differentiation [48]. These effects involve the modulation of key intracellular structures, such as focal adhesions or cell protrusions, by regulating cytoskeletal dynamics and transmitting mechanical forces throughout the cell, ultimately affecting nuclear structure and gene expression [49]. 

We designed sub-millimeter topographies on PP disks to evaluate human fibroblast responses. Roughnesses of 1.62 ± 0.14 µm and 4.16 ± 1.38 µm were obtained using patterned shot-blasted stainless steel molds, and PP substrates were fabricated by injection molding. This strategy ensured sample reproducibility, prevented contamination and residual stress within the PP, and provided cloned samples with consistent roughness, minimizing uncontrollable variables. Roughness values were chosen to provide a versatile topographical cue within the range of actin-rich structures observed in our cells, such as podosomes or stress fibers [30]. Both values exhibited significant disparity from each other, allowing a clear distinction for comparative analysis. This was particularly interesting given the variability in cell size between AW and VW fibroblasts (Appendix A), and considering that an optimal range of topographic dimensions may exist for each cell type and characteristic size [50].

Functionalization of PP surfaces with a mutated fragment of fibronectin, HBII-RGD, was successfully achieved as demonstrated by XPS. The XPS measures the atomic percentage of elements at the surface of a material, with a resolution of 10 nm in depth, hence providing information on the molecules attached at the surface of a material and corroborating each of the functionalization steps. The functionalization via silanization was properly attained on smooth and both rough PP surfaces. Oxygen plasma treatment on PP facilitates the formation of -OH (hydroxylic groups) on the material surface [39,51]. Thus, the application of the optimized plasma activation time prior to the silanization process enhanced surface wettability [39], increasing the abundance of -OH groups on the surface. These groups were key for incorporating APTES molecules and the connection with HBII-RGD. XPS measurements showed 13.07 ± 0.13% N 1s on smooth PP, a value very close to that previously obtained with the same HBII-RGD fragment on smooth Ti surfaces (14.41 ± 0.29% N 1s) [44]. Surfaces with micro-roughness exhibited slightly lower % N 1s in XPS analyses (11.85 ± 2.03% N 1s for Ra = 4.16 ± 1.38 µm and 10.28 ± 5.6% N 1s for Ra = 1.62 ± 0.14 µm), although no significant differences were observed between the three surfaces.

Both surface modifications applied to PP had significant effects on cell behavior. The introduction of roughness to non-functionalized PP substrates resulted in decreased cellularity for both AW and VW fibroblasts compared to smooth substrates, particularly evident at the highest Ra values. This reduction in cellularity was likely due to the reduced wettability of these substrates, as evidenced by the increase in WCA from 90–100° on smooth PP surfaces to 110–120° and >120° on low and high roughness surfaces, respectively. This observation is consistent with Wenzel’s theory that roughness on hydrophobic surfaces decreases wettability [47]. Furthermore, previous studies have shown that microstructures on PP surfaces significantly increase their WCA [52,53,54]. In addition, it is well known that low surface energies weaken cell adhesion [55]. This is further supported by studies indicating that micropatterning of highly hydrophobic substrates impedes fibroblast attachment and spreading compared to flat substrates [56,57]. 

Conversely, the presence of HBII-RGD on PP surfaces increased wettability on smooth and low-Ra substrates compared to their non-functionalized counterparts, but had no effect on those with the highest Ra. However, the effect of HBII-RGD on cellularity appeared to be primarily influenced by functionalization efficiency rather than changes in wettability. Functionalized smooth PP surfaces significantly increased the cellularity and spreading of both human fibroblast types compared to bare PP substrates. Increased cellularity was also observed on high-Ra functionalized surfaces, which was particularly significant for AW fibroblasts. In contrast, functionalization of low-Ra PP surfaces did not significantly affect cell adhesion for either fibroblast line compared to their non-functionalized counterparts. However, cell morphology tended to be more elongated on these low-Ra surfaces, suggesting improved cell-substrate adhesion. Notably, low-Ra surfaces exhibited the lowest atomic percentage of N 1s, approximately 20% lower than smooth substrates. These results suggest that while differences in wettability were observed after HBII-RGD immobilization on PP substrates, it was the amount of HBII-RGD that primarily regulated cell adhesion. Consequently, functionalized PP surfaces provided stronger adhesion to human fibroblasts compared to untreated surfaces where adhesion relied on weaker and non-specific interactions such as van der Waals forces and electrostatic interactions. These results support our previous observations that HBII-RGD functionalization similarly promotes cellularity and spreading of human mesenchymal cells and foreskin fibroblasts cultured on titanium surfaces [43,44]. Furthermore, the effects of HBII-RGD on cellularity appeared to have a threshold below which these effects were nearly undetectable in our model. 

In recent years, fibroblasts have been recognized as a highly heterogeneous cell type, exhibiting outstanding diversity both within and between different tissues [58]. Our study delved into this heterogeneity by investigating how changes in cellularity on experimental PP substrates are associated with alterations in the expression and distribution of actin and vinculin, two critical cytoskeletal proteins involved in the regulation of cell-substrate adhesion [59]. Actin forms microfilaments that provide mechanical support and mediate cell contractility, while vinculin is a component of focal adhesions that links intracellular actin to the extracellular space [60]. Previous research has shown that surface topography can influence actin dynamics and focal adhesion positioning, highlighting its critical role in modulating cell function [61,62]. Our results showed that roughness significantly affected the staining intensity of actin and vinculin in AW and VW fibroblasts compared to smooth surfaces on non-functionalized substrates. AW fibroblasts exhibited increased actin and vinculin intensity in response to roughness, especially at high Ra. Conversely, VW fibroblasts tended to aggregate into clusters that grew larger with increasing roughness, a phenomenon often associated with mesenchymal adherent cell behavior when contractile forces overcome the adhesion forces binding them to the substrate [63]. Further data showed that under standard conditions, VW fibroblasts are smaller and have thicker actin fibers compared to AW fibroblasts (Appendix A), suggesting that cell clustering may result from a combination of poor surface wettability and increased intracellular contractility induced by surface roughness. Therefore, our results suggest that roughness on PP surfaces activates cytoskeletal contractility mechanisms in both human fibroblast lines, albeit with different results. These results provide valuable insights into the complex interplay between surface topography and cellular behavior, and are consistent with previous studies demonstrating enhanced cellular contractile activity of F-actin on micropatterned surfaces, mediated by specific mechanisms responsive to microtopography curvature and geometry [64,65,66].

Furthermore, substrate functionalization with HBII-RGD slightly increased F-actin fluorescence intensity in AW fibroblasts cultured on both smooth and high-Ra substrates, whereas no significant change was observed on low-Ra substrates compared to non-functionalized surfaces. This variability in response can be attributed to differences in functionalization efficiency between substrates of different roughness. Previous studies have shown that HBII-RGD exhibits a remarkable affinity for transforming growth factor beta (TGF-β), leading to the induction of actin stress fibers and focal adhesions, as well as the upregulation of αSMA and ECM protein expression [43,44]. In contrast, actin and vinculin staining intensity decreased in VW fibroblasts on all substrates after surface functionalization. This decrease was likely due to HBII-RGD promoting spreading and substrate adhesion, thereby reducing clustering. However, VW cells cultured on functionalized high-Ra substrates exhibited irregular morphology and significantly decreased vinculin staining, indicating weaker substrate adhesion compared to AW fibroblasts. These results highlight the greater impact of roughness on VW fibroblasts despite functionalization.

Cellular and extracellular compressive forces, including those from cell-substrate interactions, affect nuclear morphology through mechanotransduction [48]. Surface microroughness has been shown to modulate chromatin architecture and cell fate by modulating the F-actin cytoskeleton [61]. In addition, Guillem-Martí et al. demonstrated that HBII-RGD interacts with integrin α5β1, the most highly expressed integrin in mesenchymal cells [44]. This interaction can propagate through the cytoskeleton and activate gene expression, thereby affecting nuclear morphology [49]. Our PP surfaces induced significant and distinct nuclear morphological changes in AW and VW fibroblasts. While AW fibroblasts exhibited minimal changes and consistent nuclear morphological parameters under the influence of surface roughness regardless of functionalization, VW fibroblasts exhibited irregularities, chromatin compaction, and smaller nuclei on both functionalized and non-functionalized substrates, highlighting the significant impact of surface roughness on these cells. 

Despite the differences, both AW and VW fibroblasts reached nearly full confluence on all experimental substrates within 6 days of seeding, allowing for qualitative assessment of viability and long-term adaptation to the substrates. Material surfaces, including PP with its low protein binding properties, interact with proteins that enhance cellular adhesion [67,68]. In addition, cells in culture can adapt gene expression and acquire new phenotypic traits in response to changing microenvironments [69], which likely contributes to the viability of human fibroblasts on all PP substrates, regardless of surface modifications.

## 5. Conclusions

The results obtained in this study confirm the covalent immobilization of the novel molecule HBII-RGD on PP surfaces with two different micro-roughness. The strategy used for the preparation of the PP surfaces was the key to compare the response of the cells to the surface modifications without adding uncontrollable variables.

Microroughness on PP decreased surface wettability, which impaired cell adhesion and promoted an apparent increase in intracellular contractility. However, the profiles differed between AW and VW human fibroblasts. Surface immobilization of HBII-RGD increased cell adhesion and the amount of cellular F-actin and vinculin compared to bare PP substrates. This is consistent with our previous results on titanium surfaces. This effect was influenced by the amount of HBII-RGD bound to the surface and the anatomical origin of the fibroblasts, which were derived from different tissues susceptible to surgical repair with PP. Therefore, we can confirm that modifications on PP were successful in differentiating the responses of human fibroblasts when in contact with these devices. Interestingly, modified PP surfaces were more challenging for VW fibroblasts in terms of cell adhesion, cytoskeletal elements, and nuclear morphology than for AW fibroblasts. This may be related to the higher complication rates observed in pelvic floor reconstruction compared to abdominal wall repair.

Smart surfaces that respond to more than one stimulus simultaneously could prove particularly valuable for biomedical applications, e.g., by allowing more effective control of cell responses. In the future, it will be interesting to refine the fabrication of PP materials with surface modifications and use them to evaluate specific responses of pathological cells. Such a model could contribute to the rational and tailored design of future materials for soft tissue repair.

## Figures and Tables

**Figure 1 polymers-16-00667-f001:**
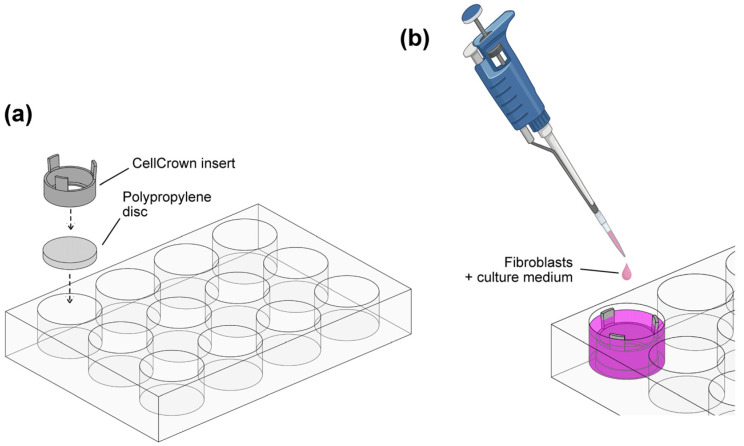
Cell culture system on polypropylene substrates. (**a**) Polypropylene discs were fixed in 12-well plates using CellCrown^TM^ plastic inserts. (**b**) Human fibroblasts were seeded on PP substrates and cultured under standard conditions for 48 h and 6 d.

**Figure 2 polymers-16-00667-f002:**
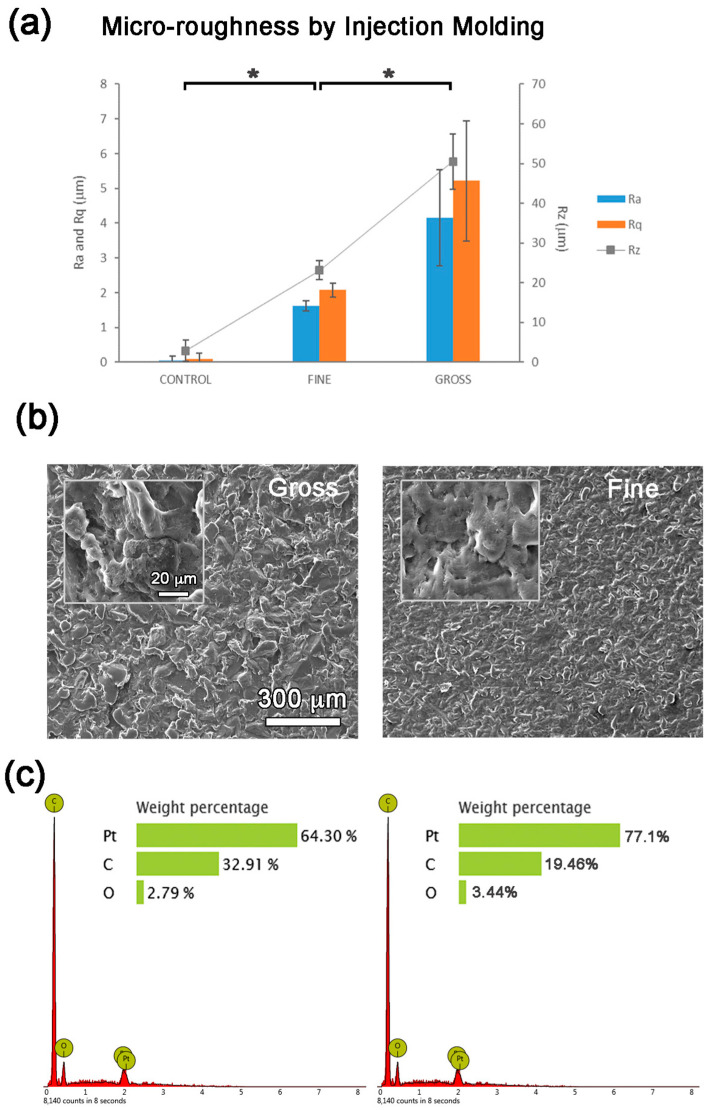
Micro-roughness analyses: (**a**) Ra, Rq and Rz roughness parameters, (**b**) SEM images and (**c**) EDS analyses of the PP surfaces prepared by injection molding. Values showing differences (*p* < 0.05) are indicated with an asterisk.

**Figure 3 polymers-16-00667-f003:**
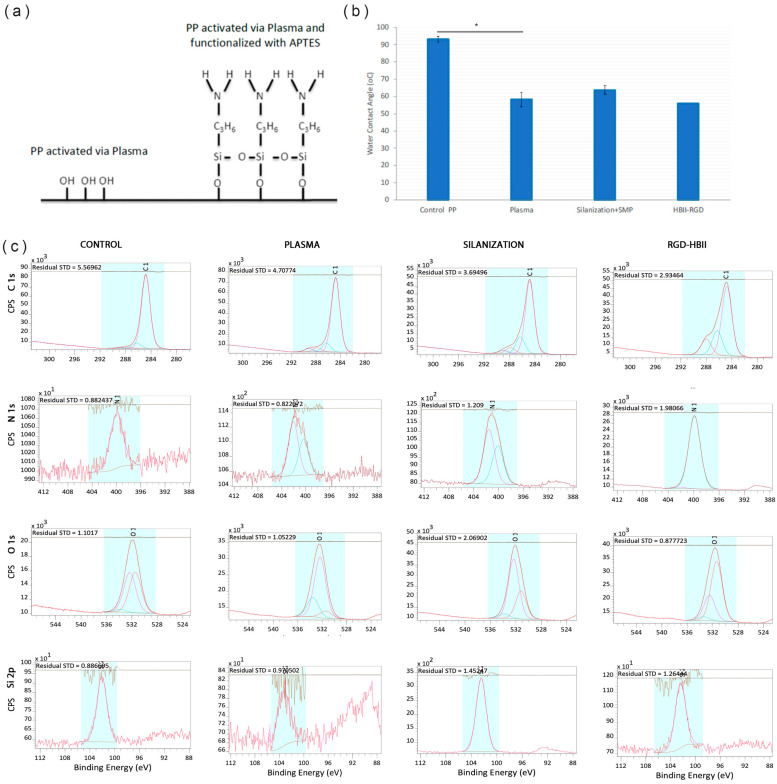
(**a**) Scheme of the silanization process. (**b**) Contact angle of the samples after activation with oxygen plasma, silanization (Silanization + SMP) and after covalent attachment of HBII-RGD to smooth PP surface (* *p* < 0.05). (**c**) Deconvolution spectra of C 1s, N 1s, O 1s and Si 2p from PP surface untreated (Control), activated with plasma oxygen (Plasma), silanized (Silanization) and functionalized with HBII-RGD (HBII-RGD).

**Figure 4 polymers-16-00667-f004:**
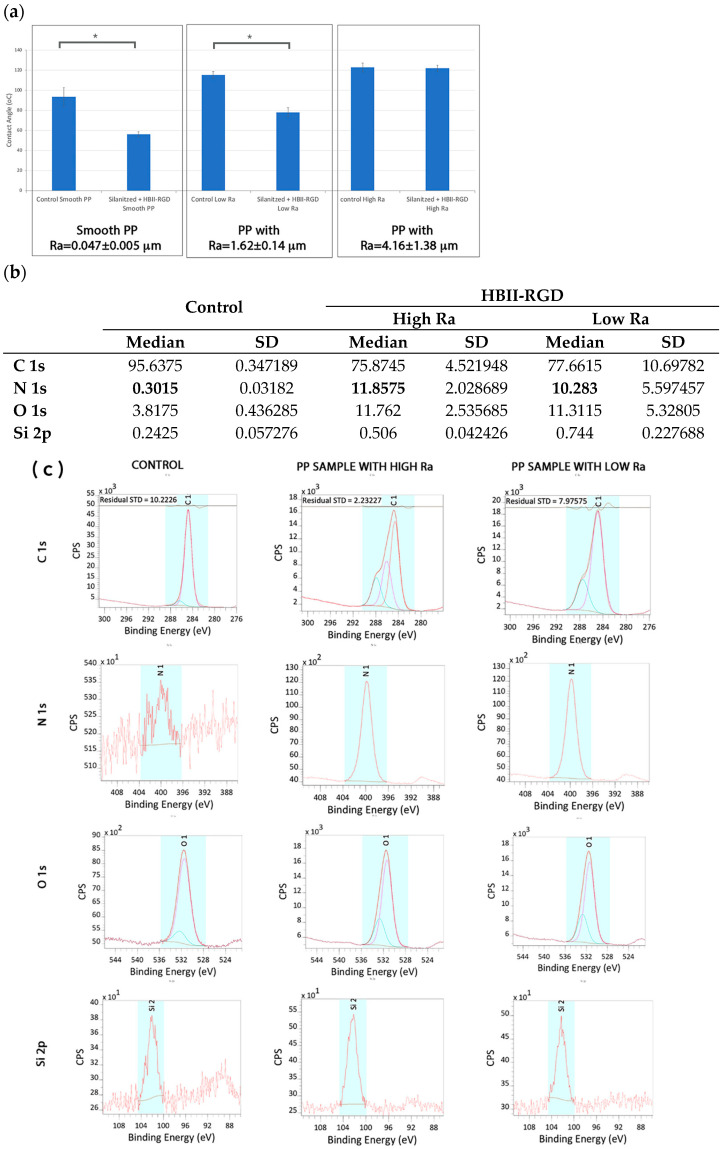
(**a**) Contact angle of PP samples before and after immobilization of HBII-RGD fragment on the surface. Values showing differences are indicated with an asterisk (*p* < 0.05). (**b**) Atomic percentage (%) distribution of the smooth untreated PP surfaces (control) and rough surfaces with low and high Ra. (**c**) Deconvolution spectra of C 1s, N 1s, O 1s and Si 2p of untreated smooth (control) and HBII-RGD functionalized rough PP surfaces (HBII-RGD; low and high Ra). The coloured peaks represent the deconvolution of each element.

**Figure 5 polymers-16-00667-f005:**
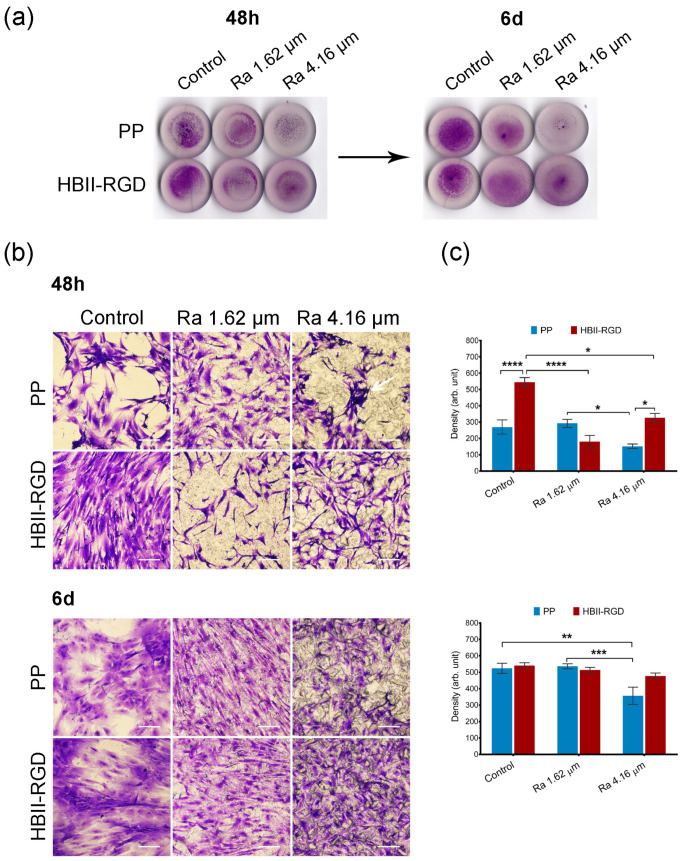
Adhesion and distribution of human AW fibroblasts on PP discs. Crystal violet staining of cells fixed 48 h and 6 d after seeding on PP surfaces with different roughness (smooth control, Ra 1.62 µm and Ra 4.16 µm), with (HBII-RGD) or without (PP) functionalization with HBII-RGD. (**a**) Representative image of entire surface of all discs analyzed after seeding. (**b**) Representative optical micrographs of the experimental surfaces stained with crystal violet dye showing cellular adhesion. Cellular debris was observed on the non-functionalized 4.16-µm Ra surface (white arrow). On functionalized rough surfaces fibroblasts showed an elongated morphology with extending projections (scale bar 50 µm). (**c**) Cell density on discs quantified by the integrated density associated with crystal violet staining (in arbitrary units) measured using Image J. Data are means ± SEM for at least three donors. Experiments were performed in duplicate. * *p* < 0.05, ** *p* < 0.01, *** *p* < 0.001, and **** *p* < 0.0001.

**Figure 6 polymers-16-00667-f006:**
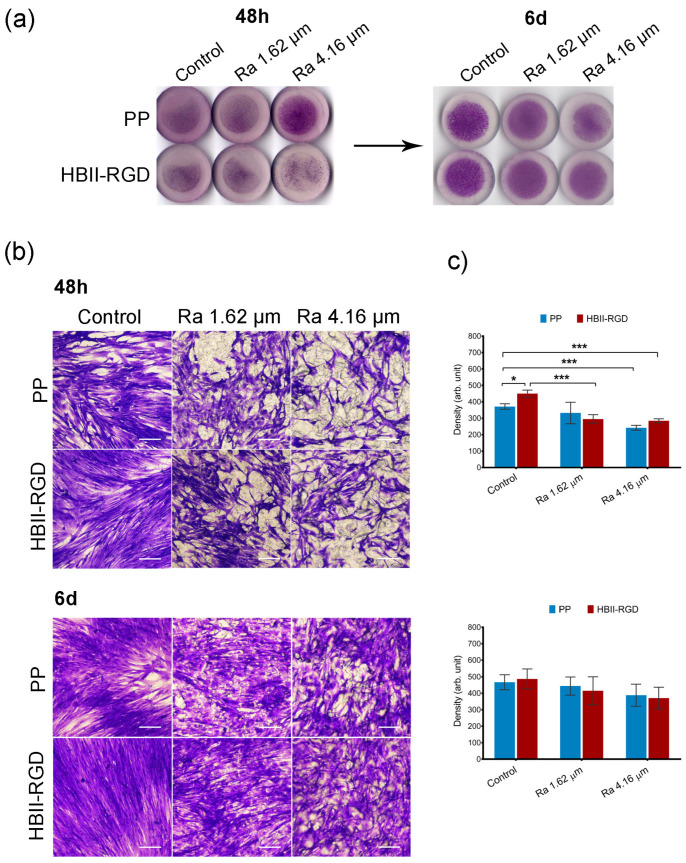
Adhesion and distribution of human VW fibroblasts on PP discs. Crystal violet staining of cells fixed 48 h and 6 d after seeding on PP surfaces with different roughness (smooth control, Ra 1.62 µm and Ra 4.16 µm), with (HBII-RGD) or without (PP) functionalization with HBII-RGD. (**a**) Representative image of the entire surface of all discs analyzed after seeding. (**b**) Representative optical micrographs of the experimental surfaces stained with crystal violet dye showing cellular adhesion (scale bar 50 µm). (**c**) Cell density on discs quantified by means of the integrated density associated with crystal violet staining (in arbitrary units) measured using Image J. Data are means ± SEM for at least three donors. Experiments were performed in duplicate. * indicates *p* < 0.05; and ***, *p* < 0.001.

**Figure 7 polymers-16-00667-f007:**
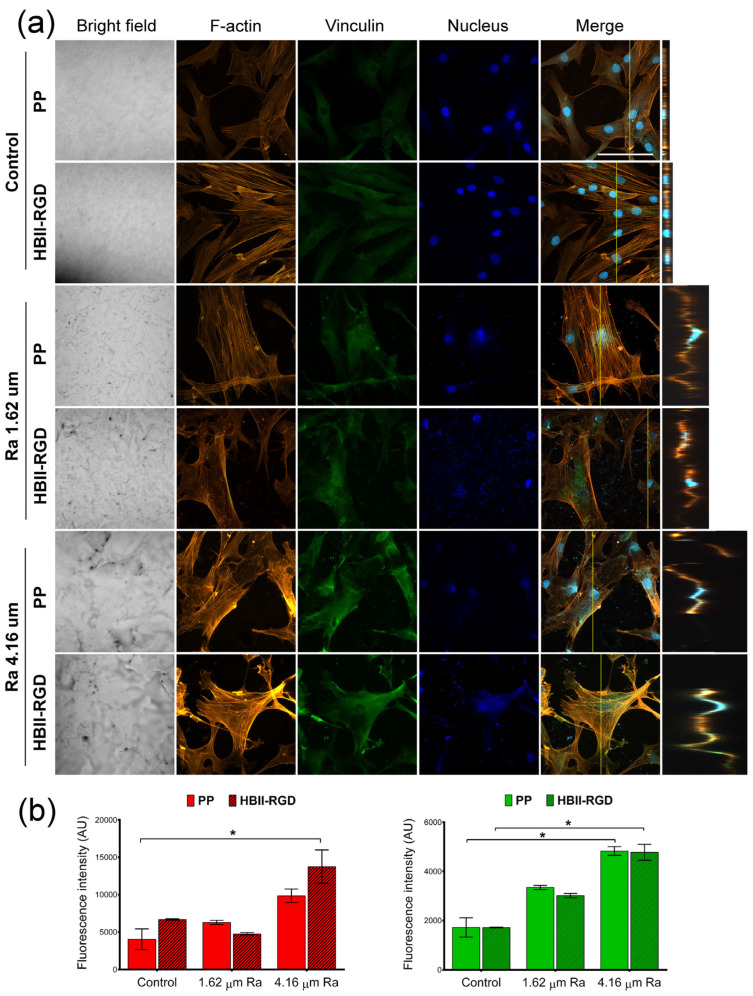
F-actin and vinculin expression in AW fibroblasts cultured on treated PP surfaces for 48 h. (**a**) Representative CLSM images of the maximum intensity projections showing F-actin (red), vinculin (green), and DNA (blue) localization (scale bar: 50 µm). Orthogonal projections along the yellow line (vertical) in the merged micrographs are shown on the right side of the panel. (**b**) Quantification of the integrated density associated with F-actin (red bars) and vinculin (green bars) fluorescence (mean ± SEM). * indicates *p* < 0.05. Control, Ra 1.62 µm, and Ra 4.16 µm indicate roughness on PP surfaces; HBII-RGD and PP indicate surface functionalization (with and without HBII-RGD, respectively).

**Figure 8 polymers-16-00667-f008:**
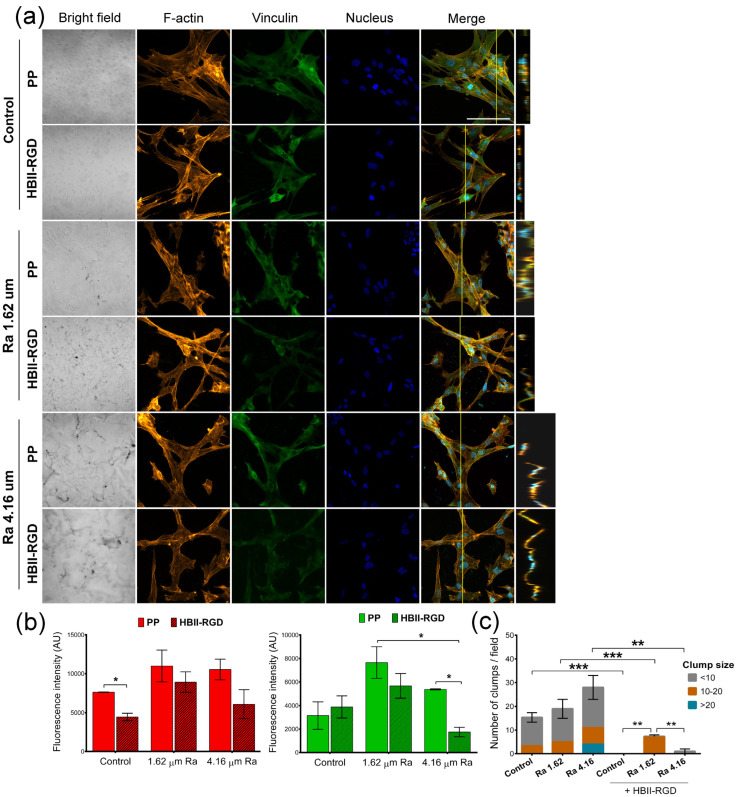
F-actin and vinculin expression in VW fibroblasts cultured on treated PP surfaces for 48 h. (**a**) Representative CLSM images of the maximum intensity projections showing F-actin (red), vinculin (green), and DNA (blue) localization (scale bar 50 µm). Orthogonal projections along the yellow line (vertical) in merged micrographs are shown on the right side of the panel. Note the formation of cell clumps (white arrow). (**b**) Quantification of the integrated density associated with F-actin (red bars) and vinculin (green bars) fluorescence (mean ± SEM). (**c**) Number and size of clumps formed after VW fibroblasts were cultured on the experimental substrates for 48 h. Bars indicate the average number and size (according to the number of nuclei) of cell clumps. * indicates *p* < 0.05; **, *p* < 0.01; and ***, *p* < 0.001. Control, Ra 1.62 µm, and Ra 4.16 µm indicate roughness on PP surfaces; HBII-RGD and PP indicate surface functionalization (with and without HBII-RGD, respectively).

**Figure 9 polymers-16-00667-f009:**
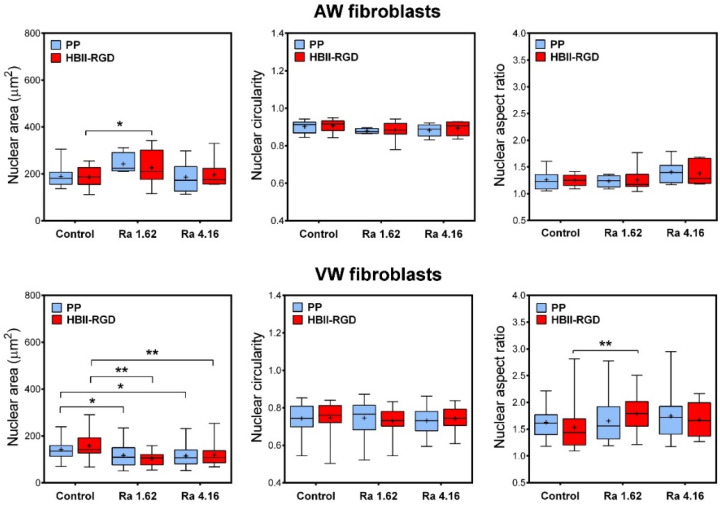
Changes in nuclear morphology of human fibroblasts cultured on the experimental PP surfaces. Box plots (median, mean, min to max) showing changes in nuclear area, circularity and aspect ratio of AW and VW fibroblasts cultured for 48 h on the experimental surfaces as assessed by fluorescence images of DAPI-stained nuclei. * indicates *p* < 0.05; **, *p* < 0.01. Control, Ra 1.62 µm, and Ra 4.16 µm indicate rouhghness on PP surfaces; HBII-RGD and PP indicate surface functionalization (with and without HBII-RGD, respectively).

**Table 1 polymers-16-00667-t001:** Atomic percentage (%) of the surface elemental compositions on smooth PP surface.

	Control		Plasma		Silanization		HBII-RGD	
	Median	SD	Median	SD	Median	SD	Median	SD
**C 1s**	93.65	0.12	87.42	0.19	71.35	1.41	73.51	0.47
**N 1s**	0.45	0.21	0.88	0.13	4.66	0.33	**13.07**	0.13
**O 1s**	5.02	0.43	**11.52**	0.11	19.74	0.88	12.78	0.27
**Si 2p**	0.50	0.02	0.19	0.05	**4.25**	0.20	0.63	0.07

**Table 2 polymers-16-00667-t002:** Atomic percentages (%) of the atomic combinations (ND-No detected).

		Control		Plasma		Silanization		HBII-RGD	
		Median	SD	Median	SD	Median	SD	Median	SD
**C 1s**	C-C, C-H	90.94	0.00	81.70	0.12	73.80	0.04	62.99	0.39
	C-O	6.46	0.04	9.88	0.01	17.28	0.00	22.49	0.17
	C=O	1.75	0.04	4.55	0.00	5.88	0.08	14.52	0.22
	N-C=O, O-C=O	0.71	0.12	3.87	0.01	3.05	0.00	NP	NP
**N 1s**	(NH4)/(NR4)	0.00	0.00	57.53	3.43	55.96	3.81	1.41	0.31
	N-(C=O)-	100.00	0.00	42.47	3.43	44.04	3.81	98.59	0.31
**O 1s**	O-C, O-H	48.10	0.07	70.61	1.68	63.40	1.66	28.56	0.09
	C=O	3.16	0.06	21.65	0.68	4.56	0.58	2.35	2.50
	O-C=O, N-C=O	48.74	0.01	7.74	1.01	32.04	2.23	69.10	2.41
**Si 2p**	Si-O	100	0	100	0	100	0	100	0

**Table 3 polymers-16-00667-t003:** Atomic Percentage (%) of the atomic combinations (ND-No detected).

			Control-Smooth Ti		Ra Low		Ra High	
		Position	Median	SD	Median	SD	Median	SD
**C 1s**	C-C, C-H	284.79	95.31	0.08	73.64	4.41	68.30	21.20
	C-O	286.31	4.69	0.08	18.25	0.33	19.78	12.90
	C=O	287.80	ND	ND	17.24	8.49	17.78	0.00
	N-C=O, O-C=O	289.19	ND	ND	ND	ND	6.04	0.00
**N 1s**	(NH4)/(NR4)	402.05	ND	ND	ND	ND	ND	ND
	N-(C=O)-	400.34	ND	ND	**100**	0	**100**	0
**O 1S**	O-C, O-H	532.34	13.76	2.71	75.47	1.53	73.30	1.04
	O-C=O, N-C=O	531.36	86.24	2.71	24.53	1.53	26.70	1.04
**Si 1s**	Si-O	103.32	100	0	100	0	100	0

## Data Availability

Data are contained within the article and Appendix A.

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
