# Peer review of "Effect of Functionalization of Texturized Polypropylene Surface by Silanization and HBII-RGD Attachment on Response of Primary Abdominal and Vaginal Fibroblasts"

_polymers, 2024, doi:10.3390/polym16050667_

Round 1
Reviewer 1 Report
Comments and Suggestions for Authors
1. It is recommended to add experiments to prove that HBII-RGD has been functionalized on polypropylene materials.
2. The clarity of many pictures in the text is poor, which affects the recognition of the text in the pictures, making it unsightly and inconvenient to read. Such as Figure3(c), Figure4(a), (c)
3. What is the distribution of surface roughness of polypropylene material during injection molding by controlling the mold? And why were the textures with two roughnesses of 1.62±0.14µm and 4.16±1.38µm chosen for analysis?
4. For Figure 4 and later, please display the error bars of the histogram completely and do not cover the lower half of the error bars.
5. Part of the discussion is not concise enough and the logic needs to be improved.
6. English grammar needs to be polished and expression needs to be improved.
Comments on the Quality of English LanguageModerate editing of English language required
Author Response
First of all, thank you for taking the time to review the manuscript. We have already revised the manuscript, taking into account your comments.
1. It is recommended to add experiments to prove that HBII-RGD has been functionalized on polypropylene materials.
Thank you for the comment and we apologize for not being clear enough in our first submission. To demonstrate the correct functionalization of HBII-RGD, we performed an XPS experiment on polypropylene materials in each of the functionalization steps (Table 1). XPS measures the atomic percentage of elements at the surface of a material with a resolution of 10 nm in depth, providing information about the molecules attached to the surface of a biomaterial. As expected, the polypropylene samples were composed of C and O. Plasma treatment resulted in an increase of O, mainly in the form of OH. Silanization resulted in a slight increase in the presence of N, due to the presence of amine groups in APTES. This increase was significantly higher when HBII-RGD was added, due to the presence of N in the protein, demonstrating the presence of HBII-RGD at the end of the functionalization steps. It is worth noting that this percentage is similar to previous work where we functionalized the HBII-RGD on Ti [1]-[2], thus demonstrating that the protein was functionalized on polypropylene. In our previous study on Ti [2], we were able to calculate the thickness of the HBII-RGD coating as a reduction of the Ti signal in XPS (approximately 5 nm). However, in our current work, the substrate, polypropylene, is composed of C as the protein, so we cannot calculate the thickness of the coating as a reduction of the C signal. However, since the percentages of N are similar, we can assume that the thickness of the coating is similar to that found in our previous work.
1 [Guillem-Marti, J.; Gelabert, M.; Heras-Parets, A.; Pegueroles, M.; Ginebra, M.P.; Manero, J.M. RGD Mutation of the Heparin Binding II Fragment of Fibronectin for Guiding Mesenchymal Stem Cell Behavior on Titanium Surfaces. A.C.S. Appl. Mater. Interfaces. 2019, 11, 3666-3678. doi: 10.1021/acsami.8b17138.]
2 [Heras-Parets, A.; Ginebra, M.P.; Manero, J.M.; Guillem-Marti, J. Guiding Fibroblast Activation Using an RGD-mutated Heparin Binding II Fragment of Fibronectin for Gingival Titanium Integration. Adv. Healthcare Mater. 2023, 12 (21), e2203307. doi: 10.1002/adhm.202203307.]
Accordingly, we have changed the main text to clarify this issue as follows:
Results section 3.3 - The samples silanized with APTES showed the presence of Si and N as Si-O and amine groups. This result confirms that the first step of the silanization process occurred correctly because the silane is forming a bond with the OH in the plasma-activated PP surface. Also, the presence of APTEs is demonstrated by the increase in the N signal because APTES contains amine groups. We observed a further increase in the N signal when the HBII-RGD fragment was added as a result of the presence of amine groups in the protein, demonstrating the correct functionalization of HBII-RGD onto the PP surface.
Discussion section - Functionalization of PP surfaces with a mutated fragment of fibronectin, HBII-RGD, was successfully achieved as demonstrated by XPS. The XPS measures the atomic percentage of elements at the surface of a material, with a resolution of 10 nm in depth, hence providing information on the molecules attached at the surface of a material and corroborating each of the functionalization steps. The functionalization via ......
2. The clarity of many pictures in the text is poor, which affects the recognition of the text in the pictures, making it unsightly and inconvenient to read. Such as Figure 3(c), Figure 4(a), (c)
The resolution of the images in Figure 3(c) and Figure 4(a and c) has been changed for better resolution.
3. What is the surface roughness distribution of polypropylene material during injection molding by controlling the mold?
The roughness (Ra) distribution on the mold as well as on the injected sheets is homogeneous because the mold was prepared by maintaining the blasting parameters: projection distance, blasting pressure, angle of incidence, and the granulometry of the blasting sand (corundum). We determined the roughness of the mold surfaces as well as the injected PP surfaces by measuring the Ra at different points on the surfaces using confocal measurements. The Ra values of both the mold and the specimens showed low standard deviations.
And why were the textures with two roughnesses of 1.62±0.14µm and 4.16±1.38µm chosen for analysis?
Previously, we and others have noted unique cytoskeletal features in fibroblasts from abdominal fascial and vaginal wall tissues affected by hernias and prolapse, respectively, compared to those from healthy tissues, which may potentially contribute to the observed phenotypic changes in these dysfunctional fibroblasts. (Diaz R 2011, 2641; Quiles M 2022, 100244).
Surface modification of materials can affect specific physical, chemical, and biological or genetic properties in various bioanalytical settings. In particular, features on material surfaces at the micrometer scale have been shown to significantly affect various cellular properties (such as cell morphology, substrate adhesion and migration, through effects on the orientation and formation of stress fibers -similar in size to the cell body-). Of interest, micropatterned substrates have been shown to alter cytoskeletal structure (i.e., actin stress fibers and myosin contractility), mostly at micrometer spacings comparable to those in our study (3-5 um) (Azatov M 2017, 065003). These cytoskeletal changes can affect nuclear morphology, chromatin condensation, and ultimately gene expression, thereby influencing cell phenotype and differentiation (Carthew J 2021, 2003186; Sun Z 2016, 215).
Therefore, our goal was to modify the surface of polypropylene (PP) at the micrometer scale to study its effects on the cytoskeleton, cell adhesion, and viability in healthy fibroblasts derived from abdominal and vaginal wall tissues. We envisioned this as a first step toward a better understanding of the interactions between these cells and PP surfaces. We chose two roughness levels that were statistically different but within the size range of actin-rich structures present in our cells, such as podosomes or stress fibers, taking into account the different sizes observed between abdominal wall fascia and vaginal wall fibroblasts (as shown in Supplementary Figure 1). We hypothesized that the different phenotypes of these fibroblasts might lead to different responses to different microtopographies.
In an attempt to better clarify this aspect, the following paragraph was added to the Discussion section: "The roughness values were chosen to provide a diverse topographic signal that falls within the spectrum of actin-rich structures observed in our cells, such as podosomes or stress fibers [30]. Both values showed notable differences from each other, allowing a clear differentiation for comparative study. This was particularly intriguing given the differences in cell size between abdominal wall (AW) and vaginal wall (VW) fibroblasts, and the possibility that there may be an ideal range of topographic dimensions for each cell type and characteristic size".
4. For Figure 4 and later, please display the error bars of the histogram completely and do not cover the lower half of the error bars.
The error bars of the graphs have been changed.
5. Part of the discussion is not concise enough and the logic needs to be improved.
We have completely revised and rewritten the discussion section (as noted in the manuscript), trying to follow your suggestions to improve conciseness and logical coherence. We are confident that the changes reflect a substantial improvement. However, we remain open to additional suggestions if necessary (in which case we would be grateful if you could let us know what specific points you feel need to be revised).
6. English grammar needs to be polished and expression needs to be improved.
We have also tried to review and improve English grammar and spelling.
Reviewer 2 Report
Comments and Suggestions for Authors
In this work, the polypropylene (PP) as a matrix was used to create microscale surfaces w/wo functionalization with an HBII-RGD molecule, a fibronectin fragment modified to include an RGD sequence for promoting cell attachment and differentiation. The results indicated that the covalent immobilization of the novel molecule HBII-RGD on PP surfaces with two different roughnesses. The proposed model is effective and provides a new framework to inform the smart material design to treat clinically compromised tissues. The study is very interesting. This paper is thorough, well-structured and likely of interest to readers focused on the study of primary abdominal and vaginal fibroblast response. In my opinion, this article can be accepted after minor revision for publication in Polymers. Before publication, it is necessary to clarify following points.
Comments
1. In Figure caption, the statistics ‘p’ is italics and should be revised to be ‘p’. In addition, in ‘2.7 statistics’ part, it should be added the sentence such as ‘Statistical significance was represented by *P < 0.05, **P < 0.01, ***P < 0.001 and ns = no significance’.
2. The ‘introduction’ part should be supplemented with more background information. For example, it was shown from second paragraph that surgical meshes, especially those made of polymers such as polypropylene (PP). In this paragraph, other materials should be added and compared. Author should discuss in detail what the advantage is for PP?

Minor editing of English language required.
Author Response
The authors would like to thank the reviewer for taking the time and consideration to review our study. Below, we attempt to address the reviewer's comments.
1. In Figure caption, the statistics ‘p’ is italics and should be revised to be ‘p’. In addition, in ‘2.7 statistics’ part, it should be added the sentence such as ‘Statistical significance was represented by *P < 0.05, **P < 0.01, ***P < 0.001 and ns = no significance’.
The figure captions have been corrected and the sentence has been added to the M&M in Section 2.7.
2. The ‘introduction’ part should be supplemented with more background information. For example, it was shown from second paragraph that surgical meshes, especially those made of polymers such as polypropylene (PP). In this paragraph, other materials should be added and compared. Author should discuss in detail what the advantage is for PP?
We appreciate the opportunity to improve the clarity and depth of the introductory section. We've revised the introduction to include a broader range of materials for surgical repair of IH and POP, as well as to emphasize the benefits of polypropylene (PP). PP's superior tensile strength, biocompatibility, and minimal tissue reactivity make it the preferred choice. We've also added three additional recent references (#33-35) that we believe may be of help to support our points.
Reviewer 3 Report
Comments and Suggestions for Authors
Dear Authors,
Congratulations on your valuable work. Here are my comments:
1. At some sentences typos, missing letters and extra spaces can be found.
2. Table 1. label should be on the same page as the table.
3. How do you explain that the contact angle decreased between the untreated PP? Why does not influence the wettability of the surfaces?
Author Response
The authors would like to thank the reviewer for his helpful and kind evaluation of our work in this study. In the following, we try to respond to his comments in order to improve the manuscript.
1. At some sentences typos, missing letters and extra spaces can be found.
The entire manuscript has been proofread and the typos have (hopefully!) all been corrected.
2. Table 1. label should be on the same page as the table.
Thank you for your observation. Following your indications, it has been corrected.
3. How do you explain that the contact angle decreased between the untreated PPP? Why does not influence the wettability of the surfaces?
If the question in the first paragraph of section 3.3: If the untreated is compared to the plasma treated, it is because of the increase in OH groups due to the plasma application. This is explained in section 3.2.
If the question is referred to compare the contact angle of untreated surfaces with the roughness: the surface roughness will increase the wettability and therefore decrease the CA [46].
Round 2
Reviewer 1 Report
Comments and Suggestions for Authors
The author has responded to my comments and made revisions to the manuscript.
Comments on the Quality of English LanguageMinor editing of English language required.
Author Response
We have reviewed the English of the manuscript.
